# Insulator-based dielectrophoresis-assisted separation of insulin secretory vesicles

**Mahta Barekatain[1†], Yameng Liu[2†], Ashley Archambeau[1], Vadim Cherezov[1], Scott Fraser[3], Kate L White[1]\*, Mark A Hayes[2]**

[1]Department of Chemistry, Bridge Institute, USC Michelson Center for Convergent Bioscience, University of Southern California, Los Angeles, United States; [2]School of Molecular Sciences, Arizona State University, Tempe, United States; [3]Department of Biological Sciences, Bridge Institute, USC Michelson Center for Convergent Bioscience, University of Southern California, Los Angeles, United States

**Abstract** Organelle heterogeneity and inter-organelle contacts within a single cell contribute to the limited sensitivity of current organelle separation techniques, thus hindering organelle subpopulation characterization. Here, we use direct current insulator-based dielectrophoresis (DC-iDEP) as an unbiased separation method and demonstrate its capability by identifying distinct distribution patterns of insulin vesicles from INS-1E insulinoma cells. A multiple voltage DC-iDEP strategy with increased range and sensitivity has been applied, and a differentiation factor (ratio of electrokinetic to dielectrophoretic mobility) has been used to characterize features of insulin vesicle distribution patterns. We observed a significant difference in the distribution pattern of insulin vesicles isolated from glucose-stimulated cells relative to unstimulated cells, in accordance with maturation of vesicles upon glucose stimulation. We interpret the difference in distribution pattern to be indicative of high-resolution separation of vesicle subpopulations. DC-iDEP provides a path for future characterization of subtle biochemical differences of organelle subpopulations within any biological system.

**\*For correspondence:**
katewhit@usc.edu

[†]These authors contributed equally to this work

## Editor's evaluation

This important study presents an exciting new method for separating organelles in an unbiased way and applies this method to the separation of distinct subpopulations of insulin vesicles. Solid evidence is presented that this method is capable of separating distinct subpopulations of insulin vesicles, but the identification of these subpopulations is incomplete and the biological significance of the proposed changes in vesicle populations remains unclear. This work will be of interest to cell biologists studying a variety of organelles.

## Introduction

Cell-to-cell heterogeneity of function within a given cell type arises from differences in genomic, epigenomic, transcriptomic, and proteomic (*Goldman et al., 2019*) components and their subcellular localizations. Some examples are subpopulations of organelles as identified for synaptic vesicles (*Crawford and Kavalali, 2015*), mitochondria (*Aryaman et al., 2018*), and insulin secretory vesicles (*Suckale and Solimena, 2010*), among others. The biochemical composition of each of the different subpopulations of organelles must vary, reflecting differences in age, maturation, or specific biological role. To better understand cell function, the organelle subpopulations must be identified so their distinct functional roles may be determined. This fundamental aspect of cell biology, quantification of organelle subpopulations, is limited by lack of appropriate experimental methods. To address this

gap in technology, we have developed a new approach for organelle subpopulation identification and quantification.

The new approach is centered on high-resolution separations technology built on conventional separation methods which are useful for numerous biological analyses. A traditional workflow consists of iterative centrifugation steps (*Harford and Bonifacino, 2011*) to isolate populations of targeted organelles for follow-up analysis with mass spectrometric or other omics-scale assays (*Huber et al., 2003*). Gradient columns have successfully isolated clathrin-coated vesicles (*Girard et al., 2005*), lipid droplets (*Brasaemle and Wolins, 2016*), mitochondria, and endoplasmic reticulum (ER) populations (*Bozidis et al., 2007*), from other organelles, based on differences in organelle densities. However, subpopulations of a given organelle with similar buoyant densities are often hard to differentiate and separate in these columns. Immunoisolation of organelles, although yielding pure populations, is limited by the need for specific organelle markers (*Lange et al., 2000*). Electrophoretic approaches, such as free-flow electrophoresis, have been adopted for use in microfluidic separation of organelles (*Lu et al., 2004*), enabling downstream analyses of the enriched fractions in the absence of major contaminants. However, these separations are limited to differentiation based on charge and size of the components. Although powerful, each of these traditional isolation approaches lacks sufficient sensitivity and robustness to isolate subpopulations of an organelle for biochemical characterizations such as proteomic analysis or imaging, except for smooth versus rough ER (*Lee et al., 2015*). Thus, new approaches which are sensitive to complex features beyond differences in density, size-to-charge or epitope recognition are needed.

One attractive strategy is to exploit higher order electric field effects which directly probe the nuanced physical properties of small complex bioparticles—differentiating on radius, zeta potential, permittivity, interfacial polarizability, charge distribution, deformability, and conductivity of the particles, to name but a few (*Chen et al., 2009*; *Hilton et al., 2020*; *Pethig, 2019*; *Matyushov, 2019*; *Hayes, 2020*). Direct current insulator-based dielectrophoresis (DC-iDEP) utilizes these features for separating subpopulations of bioparticles such as viruses, bacteria, organelles, and proteins (*Hayes, 2020*; *Ding et al., 2016*; *Jones et al., 2015*; *Liu and Hayes, 2021*). This approach offers a wide dynamic range, as it has been specifically used for separations ranging from neural progenitors and stem cells (*Liu et al., 2019*) to resistant versus susceptible strains of cellular pathogens (*Hilton et al., 2020*; *Jones et al., 2015*). In DC-iDEP, subtle biophysical differences can be distinguished by the differences in dielectrophoretic (DEP) and electrokinetic (EK) forces that result from a rich set of distinguishing factors such that all constituents of the bioparticle influence the potential for separation. Unlike epitope recognition strategies, DC-iDEP is well suited for assaying subpopulations of organelles, where differentiating factors are not known. DC-iDEP can be used as a discovery-based approach to interrogate a broader spectrum of organelle subpopulations (*Liu et al., 2019*) because the bioparticle separations occur quickly and require small sample volume (*Kim et al., 2019*; *Lapizco-Encinas and Rito-Palomares, 2007*). Here, we demonstrate the power of DC-iDEP in organelle separation, by using it to investigate subtle differences in the subpopulations of insulin vesicles upon differential stimulation in the INS-1E insulinoma model of the pancreatic β-cells (*Merglen et al., 2004*).

Pancreatic β-cells are responsible for secreting insulin in a tightly regulated process that is key to maintaining glucose homeostasis. Insulin vesicles undergo a complex functional maturation process that is required for proper secretion of insulin and this process is dysregulated in diabetes (*Suckale and Solimena, 2010*). Immature insulin vesicles act as a sorting compartment (*Feng and Arvan, 2003*; *Huang and Arvan, 1994*) and mature into two distinct pools of functional vesicles within the cell: the readily releasable pool and the reserve pool (*Boland et al., 2017*; *Orci, 1985*; *Orci et al., 1986*; *Dean, 1973*; *Figure 1A*). The heterogeneity among subpopulations of insulin vesicles (*Hou et al., 2009*; *Michael et al., 2006*; *Zhang et al., 2020*) likely arises from the varying stages of the maturation process. This accounts for modifications of insulin vesicle membrane proteins, and variances in their age, mobility, and localization within the cell (*Suckale and Solimena, 2010*; *Hao et al., 2005*; *Halban, 1982*; *Gold et al., 1982*). However, the associated biochemical constituents of insulin vesicles have remained elusive in the absence of sensitive isolation approaches. The importance of insulin vesicles in glucose homeostasis has led several groups to attempt to isolate and characterize the insulin vesicles (*Thurmond, 2007*; *Hutton et al., 1982*). While these studies and follow-up proteomics analysis of the isolated vesicles *Schvartz et al., 2012*; *Li et al., 2018*; *Brunner et al., 2007*; *Hickey et al., 2009* have provided insights into the biochemistry of these vesicles, there's very little overlap in protein IDs

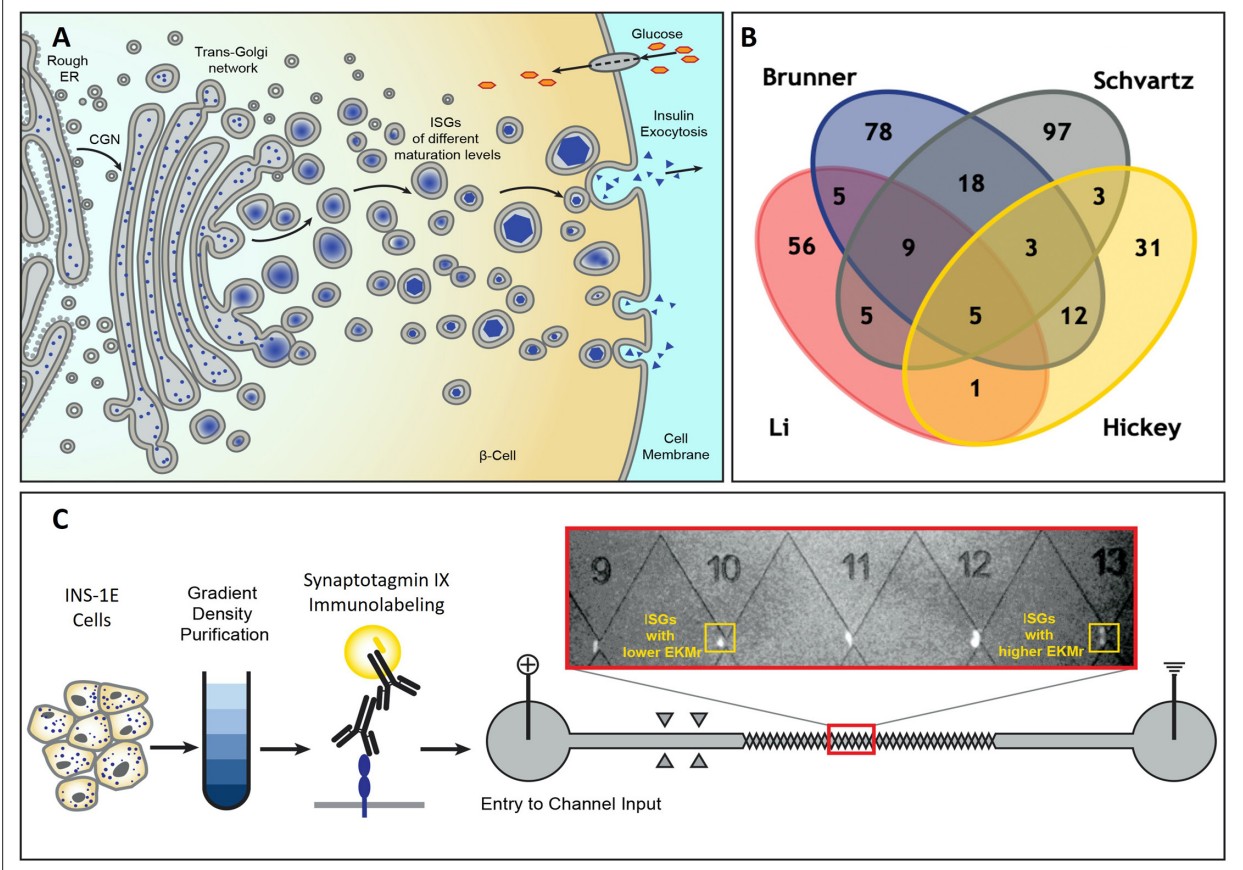

**Figure 1.** Schematic diagram for the formation of heterogenous insulin vesicles in INS-1E cells, graphical summation of disparate vesicle protein identifications, and processing of insulin vesicles including direct current insulator-based dielectrophoresis (DC-iDEP) device. (**A**) Insulin vesicle formation and maturation in a pancreatic β-cell. Newly synthesized insulin is packed inside secretory vesicles which mature to store crystalline insulin in vesicles until secretion is stimulated through different signaling pathways. (**B**) Four published insulin vesicle proteomics studies (***Schvartz et al., 2012***; ***Li et al., 2018***; ***Brunner et al., 2007***; ***Hickey et al., 2009***) aimed to identify the proteome of the heterogenous populations of secretory vesicles in INS-1E cells with only five proteins identified consistently. (**C**) Separation of insulin vesicles using a DC-iDEP device. Differential and density gradient centrifugation were used to enrich each sample for insulin vesicle populations. Samples were then immunolabeled and introduced into DC-iDEP device for high-resolution separation. Fluorescently labeled particles trapped near various gates in the channel are biophysically different subpopulations with varied EKMr values. The gates were constricted by increasing sizes of paired triangles, forming channel widths of 73 µm to 25 µm from inlet to the outlet. The different gates created $\vec{E}$ and $\nabla\|\vec{E}\|$ distributions for EKMr values.

The online version of this article includes the following source data and figure supplement(s) for figure 1:

**Figure supplement 1.** Glucose sensitivity of INS-1E insulinoma cells was tested by stimulation at increasing concentrations of glucose and measurement of insulin secretion by enzyme linked immunosorbent assay (ELISA).

**Figure supplement 2.** Selection of fractions for use in separation experiments.

**Figure supplement 2—source data 1.** Full image of western blotting (WB) with labels indicating synaptotagmin IX, SEC16 B, and cytochrome *c*.

**Figure supplement 2—source data 2.** Raw image of western blotting (WB).

**Figure supplement 3.** The identity of insulin vesicles was confirmed via several microscopy methods: confocal microscopy, transmission electron microscopy, and cryo-electron microscopy.

associated with these organelles from different studies (***Figure 1B***). Thus, there is a need for more robust isolation methods that can reproducibly differentiate between the heterogeneous subpopulations of insulin vesicles, among other organelles, and allow for their downstream characterization.

In this study, a new scanning voltage DC-iDEP separation strategy has been applied to immunolabeled insulin vesicles of the INS-1E insulinoma cells and has been shown to separate the full range of insulin vesicle subpopulations with improved resolution within multiple ranges of biophysical parameters (***Figure 1C***). We demonstrate that glucose treatment, which has been shown to influence the maturity or molecular content (***White et al., 2020***; ***Loconte et al., 2022***), affects the biophysical

characteristics associated with the vesicle subpopulations captured within our DC-iDEP device. This method allows discovery of subpopulations with distinct biophysical properties among insulin vesicles from untreated cells (n-insulin vesicles) and 25 mM glucose-treated cells (g-insulin vesicles). Our observations are consistent with previous studies where a pronounced shift in the molecular density of the insulin vesicles was noted under the two conditions (*White et al., 2020*; *Loconte et al., 2022*). This study substantiates the sensitivity of DC-iDEP separation technique in resolving subpopulations of insulin vesicles, among other organelles, and opens the avenue for numerous studies of the biochemical constituents where complex and heterogeneous populations of organelles are of interest.

## Results
### Analysis of enriched insulin vesicle samples post fractionation
The INS-1E cells used in these studies were capable of insulin secretion in response to increasing concentrations of glucose (*Figure 1—figure supplement 1*). Isolated membrane fractions from differential and density gradient centrifugations were analyzed through enzyme linked immunosorbent assay (ELISA), western blotting (WB), dynamic light scattering (DLS), confocal microscopy, and electron microscopy (EM). ELISAs identified that lower fractions of the density column (9–12) contained the highest insulin content (*Figure 1—figure supplement 2A*), the same fractions shown to be enriched in the insulin vesicle marker, synaptotagmin IX, by WB (*Figure 1—figure supplement 2B*). DLS indicated that these fractions contained particles of 150–200 nm radius, which corresponds to the known range of insulin vesicles radii (*Olofsson et al., 2002*; *Greider et al., 1969*). Although the WB revealed that the final enriched vesicle sample included some contaminants from unwanted organelles, such as the ER (*Figure 1—figure supplement 2B*), this will not confound the analysis of insulin vesicles in the iDEP device as the fluorescent labeling was targeted only at insulin vesicles.

To verify that the insulin vesicles were intact prior to DC-iDEP, we imaged a modified INS-1E cell line that contains a human insulin and green fluorescent protein-tagged C peptide (hPro-CpepSfGFP) (*Haataja et al., 2013*). This GFP tag allowed for quick visual verification of intact vesicles using fluorescence confocal microscopy. We observed distinct puncta rather than a diffuse GFP signal which indicated that the vesicles were intact and not ruptured. Further analysis of isolated vesicles was done using EM. We observed intact vesicles with the expected size and shape using both transmission electron microscopy (TEM) and cryo-electron microscopy (cryo-EM) (*Figure 1—figure supplement 3*).

### Introduction of DC-iDEP as a discovery and quantification tool for insulin vesicle subpopulations
Enriched vesicle samples from INS-1E cells were subjected to analysis using DC-iDEP after labeling with insulin vesicle marker synaptotagmin IX (*Schvartz et al., 2012*; *Brunner et al., 2007*), which was confirmed to colocalize with insulin in fluorescence microscopy imaging (*Figure 2*). The separation system was operated in a discovery or scanning mode by first applying a high voltage (2100 V, empirically determined, *Figure 3*) such that all particles were prevented from entering the first gate, because DEP forces exceed EK forces. The voltage was then lowered incrementally (300 V for each step) allowing various subpopulations to enter the separation zone and be sorted along the device according to their specific EKMr values (*Figure 3*). A bolus forms at a gate, corresponding to an EKMr value that is a result of a balance between DEP and EK forces on each particle and reflects a complex set of biological, chemical, and biophysical properties of the vesicles (*Jones et al., 2015*; *Hilton and Hayes, 2019*; *Crowther et al., 2019*; *Liu and Hayes, 2020*). The fluorescence intensity was captured for each gate over a full range of voltages (1800–600 V), such that the largest EKMr values are probed with the higher applied voltage (*Figure 4*).

### Biophysical subpopulations of vesicles from untreated INS-1E cells
The distribution of fluorescently labeled n-insulin vesicles captured at each gate formed a characteristic arc (*Figure 1C*), indicating a well-operating and consistent system. Higher voltages provide a broader dynamic range of EKMr values for capturing a wider range of particles, while lower voltages provide detailed distribution of particles based on their associated EKMr values. At an applied voltage of 1800 V, particles were sensed at EKMr values below $1.5 \times 10^{10}$ V/m$^2$ in patterns of overlapping subpopulations (*Figure 4A*, blue circles). These overlapping features begin to spread out with an

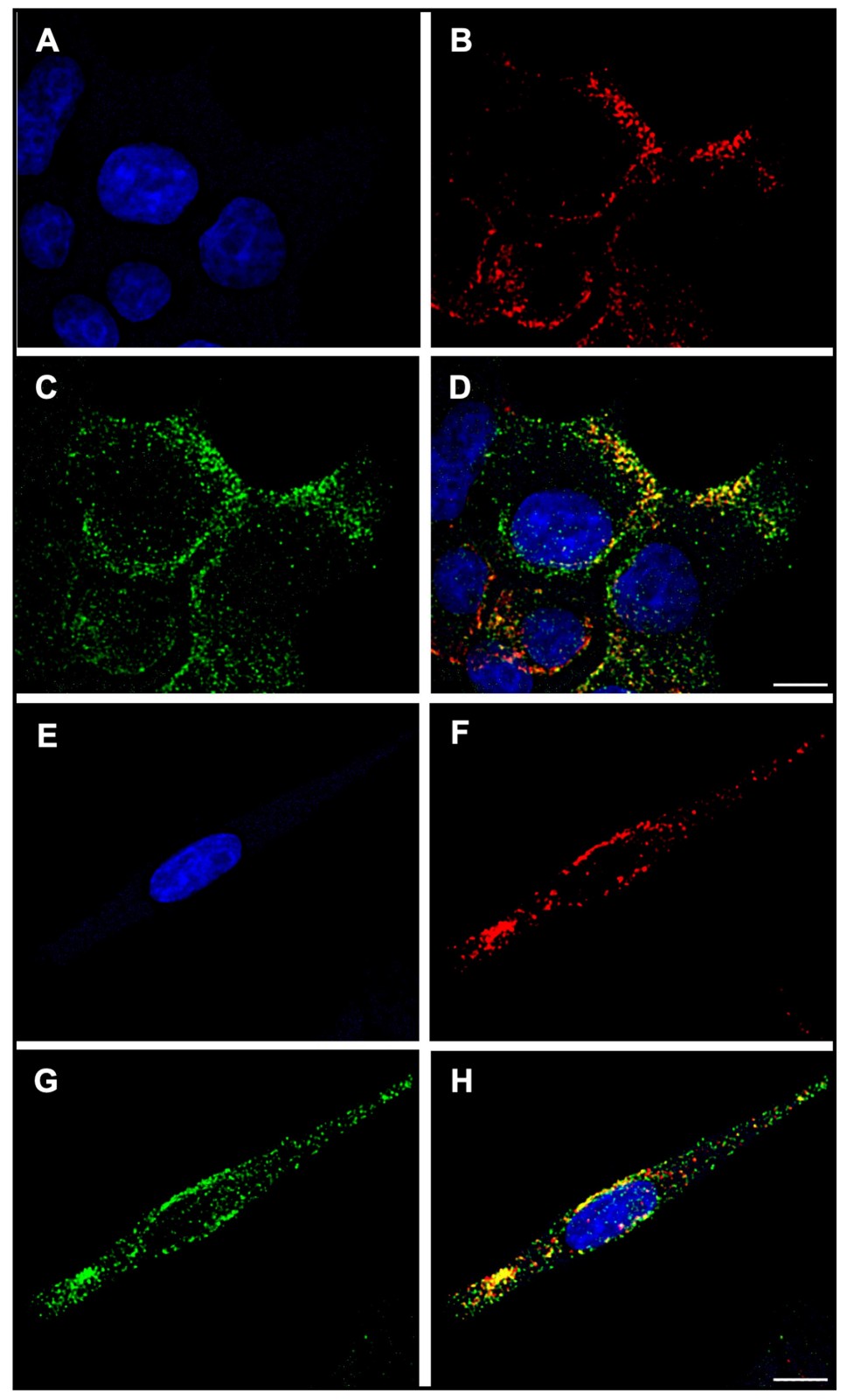

**Figure 2.** Colocalization of insulin vesicle marker used in this study (synaptotagmin IX) with insulin. (**A, E**) Nuclei of INS-1E insulinoma cells stained with NucBlue. (**B, F**) Synaptotagmin IX labeled with rabbit anti-synaptotagmin IX and goat anti-rabbit IgG (H+L), Alexa 647. (**C, G**) Insulin hormone labeled with mouse anti-insulin and goat anti-mouse IgG (H+L), Alexa 488. (**D, H**) Localization of synaptotagmin IX to insulin vesicles as apparent from the

*Figure 2 continued on next page*

*Figure 2 continued*
merged intensities of panels (**B**) and (**C**) or (**F**) and (**G**). A strong colocalization was observed between insulin and synaptotagmin IX. Pearson's r value, 0.66 (**D**) and 0.64 (**H**). Microscopy was performed with a Leica Mica using a 63×/1.2NA water immersion objective on cells mounted in ProLong Glass Antifade Mountant. Scale bars, 5 μm.

applied voltage of 1500 V. At incrementally lower settings of applied voltages (1200, 900, and 600 V), distinctive patterns become identifiable (*Figure 4B–E*). Notable and discernible features of bioparticle distribution are apparent around $1.2\times10^{10}$ and $1.8\times10^{10}$ V/m$^2$ at an applied voltage of 1200 V (*Figure 4C*). Lowering the voltage to 900 V, and redistribution of bioparticles based on adapted properties of the channel, reveals a similar but attenuated feature of the distribution around $1.2\times10^{10}$ V/m$^2$ (*Figure 4D*), whereas particles with EKMr values greater than $1.5\times10^{10}$ V/m$^2$ leave the channel at this voltage. At this voltage, redistribution of particles, previously retained in overlapping patterns at 1200 V, forms a distinct peak around $5–6\times10^{9}$ V/m$^2$ (*Figure 4D*). Further lowering the voltage to 600 V shows similar patterns of particle distribution around $5–6\times10^{9}$ V/m$^2$, as well as distinctive patterns around $3–4\times10^{9}$ V/m$^2$ (*Figure 4E*), while leaving out populations with EKMr values higher than $1.0\times10^{10}$ V/m$^2$.

## Biophysical subpopulations of vesicles from glucose-stimulated INS-1E cells

Insulin vesicles obtained from 25 mM glucose-treated INS-1E cells (g-insulin vesicles) were studied with the same method used for the untreated cells (*Figure 4*, yellow squares). Consistent with the vesicles from untreated cells, patterns of primarily overlapping subpopulations were detectable at a voltage of 1800 V. Distribution of particles was observed at values up to $2.3\times10^{10}$ V/m$^2$ (compared to a maximum value of $1.5\times10^{10}$ V/m$^2$ for the untreated populations) (*Figure 4A*), which suggests that these vesicles have a broader range of properties than the population from the untreated cells. The first evidence of a distinct distribution feature was captured around $7–8\times10^{9}$ V/m$^2$ when the voltage was lowered to 1500 V (*Figure 4B*). Decreasing the voltage to 1200 V and subsequent particle redistribution in the channel revealed a distribution pattern with discernible features around $7–8\times10^{9}$ V/m$^2$, like those observed at 1500 V, as well as a distinct peak around $1.1\times10^{10}$ V/m$^2$ (*Figure 4C*). Further

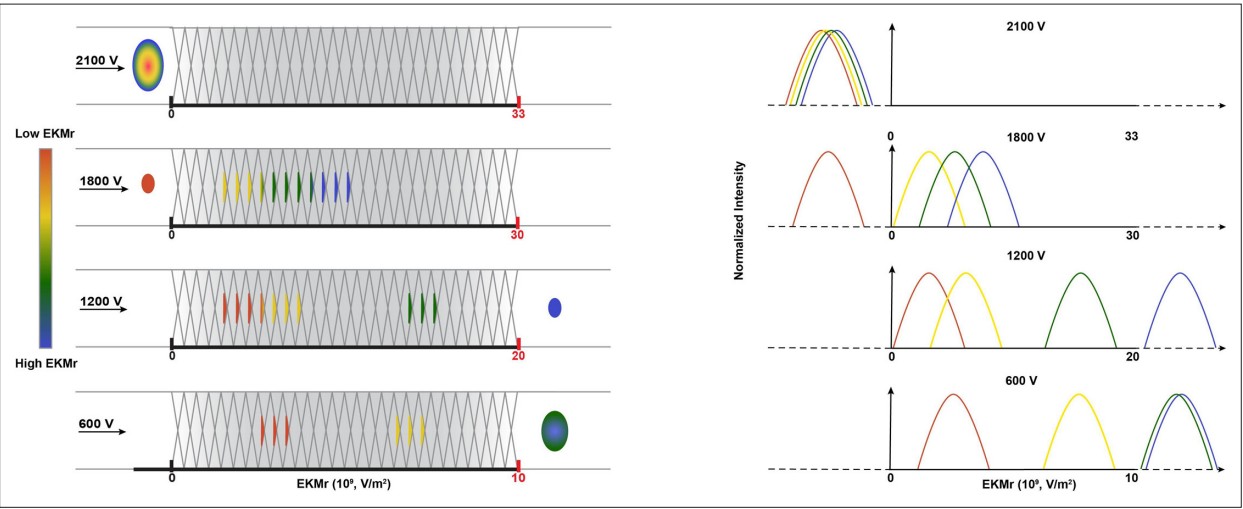

**Figure 3.** Schematic diagram of direct current insulator-based dielectrophoresis (DC-iDEP) system operated in a discovery or scanning mode. As a function of the design of the sawtooth channel with different gate sizes along the channel, the applied voltage defines dielectrophoretic (DEP) and electrokinetic (EK) forces at each gate, and capture occurs when the EK force of the particle is equal to or smaller than the DEP force. At high voltages, only particles with high EKMr values can enter the channel; the highest applied voltage of 2100 V prevents all particles of the sample from entering the inlet of the device due to the induced dielectrophoretic forces (no fluorescent signal detected anywhere along the channel). Sequentially lower voltages allow the various subpopulations to enter and be separated throughout the channel. When a subpopulation's EKMr value surpasses the channel's DEP force limit, it travels freely and leaves the channel at the outlet. The right panel indicates fluorescent intensities of the captured particles are recorded along the channel at each voltage. Tracking these intensities allows the discovery and quantification of unknown subpopulations according to their biophysical properties.

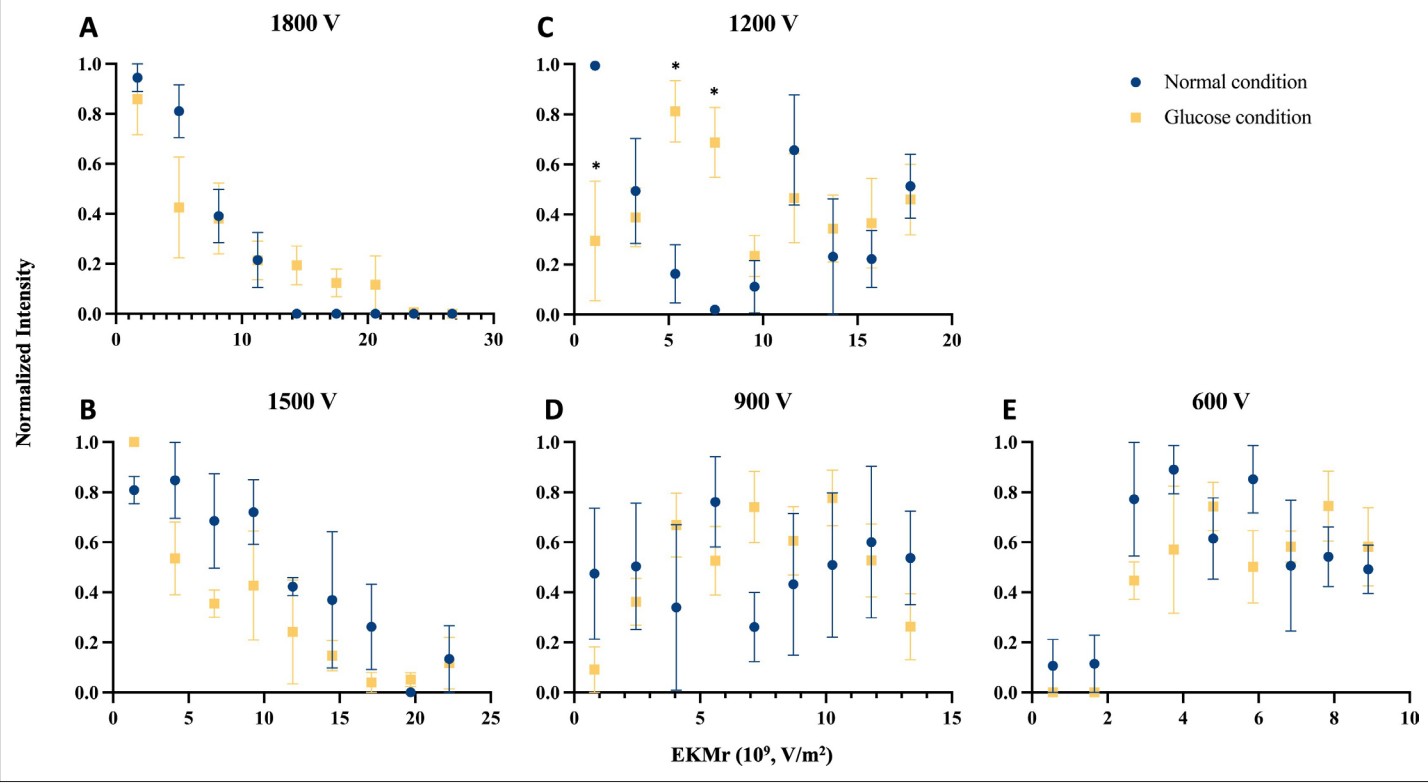

**Figure 4.** Comparison of distributions for the biophysical properties as reflected in EKMr values of n-insulin vesicles (blue circles) and g-insulin vesicles (yellow squares) with varied voltages applied. Fluorescent intensities of captured n- and g-insulin vesicles with different EKMr values were recorded at each gate. Each data point reflects fluorescent intensities recorded at three subsequent gates with the same EKMr values, averaged out over biologically replicated experiments and normalized over all signals recorded at a given voltage. (**A**) Full profile of the sample's biophysical distribution was recorded at 1800 V. (**B–E**) Insulin vesicle subpopulations were separated at subsequent applied voltages of 1500, 1200, 900, and 600 V. Values are mean ± SEM (n=3 for n-vesicles or 4 for g-vesicles for biologically independent experiments) (*p<0.05 using ANOVA with Bonferroni post hoc multiple comparison correction). Raw data in *Figure 4—source data 1*.

The online version of this article includes the following source data for figure 4:

**Source data 1.** The raw data depicted in *Figure 4*.

lowering the voltage to 900 V resulted in a unique distribution pattern with discernible features around $1.1 \times 10^{10}$ V/m² (*Figure 4D*), similar to 1200 V, and $8 \times 10^9$ V/m² (*Figure 4D*), previously observed at 1500 and 1200 V (*Figure 4B and C*). Another feature of this distribution pattern was a peak around $4 \times 10^9$ V/m² (*Figure 4D*). Ultimately, decreasing the voltage to 600 V revealed a distribution pattern with features around $4 \times 10^9$ and $8 \times 10^9$ V/m² (*Figure 4E*), like those observed at higher voltages, leaving out patterns that were observed at EKMr values higher than $1.0 \times 10^{10}$ V/m². This distinct distribution pattern, although consistent with distribution patterns at higher voltages, featured stretched-out peaks, consistent with a smaller and more refined range of EKMr values assigned throughout the channel.

## Comparison of n- and g-insulin vesicle separation patterns within the iDEP device

Visual inspection of the collected data revealed generally similar patterns of vesicles collected at specific EKMr values (*Figure 4*). However, at 1200 V we achieved adequate separation of vesicle populations to discern unique populations of vesicles from cells treated with glucose compared to no treatment. Using a two-way ANOVA, we found a statistically significant interaction between the effect of treatment on vesicles collected at each EKMr value for data collected only at 1200 V (F(8, 45)=3.61, p=0.003). A Bonferroni post hoc test revealed a significant difference in the intensity or quantity of vesicles collected between treated and untreated samples at $1.10 \times 10^9$ V/m² (p=0.0249), $5.35 \times 10^9$ V/

m$^2$ (p=0.0469), 7.45×10$^9$ V/m$^2$ (p=0.0369). These differences reflect a shift in the populations of insulin vesicles upon glucose stimulation.

## Discussion

Using the DC-iDEP separation method, we identified distinct populations of insulin vesicles with varying biophysical properties. Furthermore, we observed different distribution patterns for insulin vesicles isolated from glucose-stimulated versus unstimulated cells, which suggests glucose stimulation alters the insulin vesicle subpopulations. This is consistent with findings from single-cell soft X-ray tomography, which revealed heterogeneities in the composition of insulin vesicles and their molecular densities upon glucose stimulation due to vesicle maturation (*White et al., 2020*; *Loconte et al., 2022*). Other studies have also observed enrichment of certain subpopulations of vesicles in response to glucose stimulation (*Straub et al., 2004*). Our approach addresses an important need to identify and separate distinct subpopulations of insulin vesicles which can allow for investigating their apparent heterogeneity.

The intensity peaks we observed at specific EKMr values likely correspond to some of the previously described insulin vesicle subpopulations (*Zhang et al., 2020*; *Norris et al., 2021*; *Neukam et al., 2020*; *Yau et al., 2020*; *Kreutzberger et al., 2020*). Larger particles are expected to have a smaller EKMr value compared to smaller particles (*Hilton et al., 2020*). Subpopulations containing larger insulin vesicles, such as a mature pool (*Zhang et al., 2020*; *Norris et al., 2021*), synaptotagmin IX-positive vesicles (*Kreutzberger et al., 2020*), or docked vesicles near the plasma membrane (*Zhang et al., 2020*) may have lower EKMr values than smaller immature vesicles. Additionally, phosphatidylcholine lipids increase the zeta potential of tristearoylglycerol crystals (*Arts et al., 1994*). This effect may extend to insulin vesicle subpopulations containing more phosphatidylcholine, such as young insulin vesicles (*Neukam et al., 2020*) which could lead to higher EKMr values. Taken together, these two properties may be used to predict the EKMr values of known insulin vesicle subpopulations. For example, insulin vesicles with EKMr values of 1–2×10$^9$ V/m$^2$ (*Figure 4C*) may represent a synaptotagmin IX-positive subpopulation due to their larger radii and depletion under glucose stimulation. Additionally, young insulin vesicles may have EKMr values between 5 and 7.5×10$^9$ V/m$^2$ (*Figure 4C*) due to higher amounts of phosphatidylcholine present in this subpopulation (*Neukam et al., 2020*). In this EKMr range, we observed a higher intensity for glucose-treated cells which may suggest biosynthesis of new vesicles. Immature insulin vesicles are likely to have higher EKMr values due to their smaller size (*Zhang et al., 2020*), such as an EKMr value between 1.5 and 1.6×10$^{10}$ V/m$^2$ (*Figure 4C*). Here, we demonstrated the capabilities of DC-iDEP to separate insulin vesicle subpopulations in an unbiased manner. Future experiments using chemical probes to label subpopulations will be useful to accurately define the EKMr values associated with specific subpopulations.

The intensity at each EKMr value is influenced by individual insulin vesicles. The distribution of this pattern reflects differences in physical properties of the vesicles such as physical changes to the vesicles as they mature. The specific changes to the individual insulin vesicles which result in varied EKMr measurements can be associated with any alteration of the biochemical makeup of the bioparticle. There is an ongoing evolution of the theoretical underpinnings of electric field gradient techniques, where past physical descriptors were limited to conductivity and permittivity of the particle (*Pethig, 2019*). It is becoming better understood that the overall structure and subtle details of the bioparticle, including the particle-solvent cross-polarizations, will influence these forces (*Pethig, 2019*; *Matyushov, 2019*; *Hayes, 2020*; *Hölzel and Pethig, 2021*; *Martin and Matyushov, 2012*). This new view of the forces imparted on the insulin vesicles aligns with a rather simple proposition that the makeup of the particle has changed: something has been added, subtracted, or altered which changed the force on the particle in a measurable and quantifiable way. This is consistent with findings that glucose can affect the maturation of existing vesicles, increasing their molecular density or the concentration of biomolecules within the vesicle lumen (*White et al., 2020*). Additionally, the subpopulations we observe could have been altered by a change in surface protein expression or enrichment of unsaturated lipids (*Moore et al., 2019*). This is also consistent with findings that glucose enhances vesicle-mitochondria association which is hypothesized to contribute to insulin vesicle maturation (*White et al., 2020*). Identical particles always have the same EKMr in the same manner that identical proteins always have the same molecular weight (*Zhu et al., 2019*). Thus, we can use EKMr values to evaluate

the presence of specific insulin vesicle subpopulations and explore how specific chemical probes may influence vesicle identity.

There are some subtleties in the presented data which accentuate features, and some limitations, of the DC-iDEP separations and the imaging system. The data for 1800 V, for instance, shows no discernable local maxima above $1.5\times10^{10}$ V/m$^2$ for the n-vesicles, despite having identifiable populations with EKMr values higher than $1.5\times10^{10}$ V/m$^2$ at lower voltages. Another feature that is quite apparent is a lack of distinct 'peaks' or identifiable patterns appearing at consistent EKMr values within the datasets from differing applied voltage values. While the applied voltage does not affect the properties of bioparticles, it defines the forces that oppose bioparticle movements across the channel at different gates. Accordingly, in an overlapping pattern of subpopulations, those with higher EKMr values overcome the weaker opposing forces at a given gate once the voltage is lowered (*Liu and Hayes, 2021*; *Jones and Hayes, 2015*). Hence, the signal which was previously averaged with signals from the other overlapping subpopulations is now discernible at a higher EKMr value. Considering the high resolution of the technique, homogeneous subpopulations of vesicles undoubtably consist of a very narrow range of EKMr values. Each data point shown in *Figure 4* represents a homogenous subpopulation captured at the gate. While this feature is unsatisfying to classic separation scientists, it still allows for quantitative comparison of paired samples, as has been shown here. To further optimize the separation, one could modify the channels including the gate size and periodicity, as well as scan even more refined ranges of voltages to induce varying spectra of EKMr values along the device (*Ding et al., 2016*; *Jones et al., 2015*). Additionally, developing the capability to port the collected individual boluses will enable downstream analyses such as mass spectrometry or EM, transiting the technique from analytical to preparative.

In essence, the current work introduces DC-iDEP in a scanning mode as a powerful tool to interrogate complex organelle subpopulations and study their distinctive distribution patterns under different treatment conditions with the goal of organelle subpopulation discovery and quantification. Further, the putatively subtle differences in subpopulations between stimulated and unstimulated INS-1E insulinoma cells are quantified more extensively than previously possible. Finally, this method serves as a stepping stone toward isolation and concentration of fractions which show the largest difference between the two population patterns for further bioanalysis (imaging, proteomics, lipidomics, etc.) that otherwise would not be possible given the low-abundance components of these subpopulations. This approach can be broadly applied to any cell type and organelle beyond the scope of our model system of insulin vesicles and INS-1E insulinoma cells.

## Materials and methods

No statistical methods were used to predetermine sample size. The experiments were not randomized, and investigators were not blinded to allocation during experiments and outcome assessment.

### Cell culture

INS-1E cells (Addex Bio C0018009; RRID: CVCL_0351) were cultured according to the supplier's protocol. Briefly, cells were seeded in RPMI 1640 media (modified to contain 2 mM L-glutamine; 10 mM HEPES pH 7.2, 1 mM sodium pyruvate, 2 g/L glucose, and 1.5 g/L sodium bicarbonate, 50 μM 2-mercaptoethanol, 100 U/mL penicillin, and 100 μg/mL streptomycin; sterile filtered through 0.22 μm filter) supplemented with 10% fetal bovine serum (FBS) and grown to 80% confluency. Cells were plated at a density of $10^5$ cells/cm$^2$ in a 24-well plate for the glucose sensitivity assay and incubated at 37°C with 5% CO$_2$ in growth media for 4–5 days. Cells were pretreated at 60–80% confluency with Krebs-Ringer bicarbonate HEPES (KRBH) buffer (135 mM NaCl, 3.6 mM KCl, 5 mM NaHCO$_3$, 0.5 mM NaH$_2$PO$_4$, 0.5 mM MgCl$_2$, 1.5 mM CaCl$_2$, 10 mM HEPES, pH 7.4, and 0.1% bovine serum albumin [BSA]; made fresh within 7 days of use) free of glucose and incubated at 37°C with 5% CO$_2$ for 30 min. They were then stimulated for 30 min at 37°C using KRBH buffers containing 1.1, 5.6, 8.4, 11.1, 16.7, and 25 mM glucose, in the presence of house-made protease inhibitor (PI) cocktail (0.5 M AEBSF, 1 mM E-64, 1.13 mM leupeptin, and 151.36 μM aprotinin). KRBH buffer was removed and saved from cells for downstream analysis. Insulin secretion in response to increasing concentrations of glucose was confirmed in an ELISA (Mercodia 10-1250-01) following the manufacturer's manual (*Figure 1— figure supplement 1*). For each biological replica, cells were plated at a density of $4\times10^4$ cells/cm$^2$ in

a five-layer cell chamber (VWR 76045-402) to yield enough material for completing the experiment. All cell stacks were at least 95% viable after the harvest with 0.05% trypsin. For glucose treatment, near 80% confluent cells were gently rinsed with dialyzed phosphate-buffered saline (PBS) twice and starved in a KRBH buffer with no glucose for 30 min, followed by a 30 min stimulation of insulin release by KRBH buffer supplemented with 25 mM glucose. Cells were then harvested by mild trypsinization.

## Colocalization of synaptotagmin IX and insulin vesicles

Cells were grown on ibidi 8 well high ibiTreat slides (80806-96), precoated with poly-L-lysine. Cells were fixed with 4% ice-cold PFA for 10 min, and then stained with antibody cocktail (Mouse anti-insulin antibody [Cell Signaling Technology #8138], 1:100; Rabbit anti-synaptotagmin IX [Thermo Fisher Scientific PA5-44987; RRID: AB_2610517], 1:100) in 0.5% BSA, 0.2% saponin, 1% FBS of PBS buffer for 2 hr at room temperature (RT). After three washes with PBST for 10 min, cells were incubated for 1 hr with secondary antibody cocktail (Goat anti-Rabbit IgG (H+L) Highly Cross-Adsorbed Secondary Antibody, Alexa Fluor 488 [Invitrogen A-11034] 1:100; Goat anti-Mouse IgG (H+L) Highly Cross-Adsorbed Secondary Antibody, Alexa Fluor 647 [Invitrogen A-21236], 1:100). Cells were then mounted in ProLong Glass Antifade Mountant with NucBlue Stain (Thermo Fisher Scientific P36981) and cured for 24 hr before imaging.

A Leica Mica microhub was used for confocal imaging. Image acquisition was performed using 63×/1.2NA water immersion objective, using 888×742 format, with 0.04 µm pixel size, and pinhole diameter automatically set. The signal was collected with 359, 499, and 650 nm excitations and 461, 520, 668 nm emissions for nuclei, insulin, and synaptotagmin IX by HyD FS detector, respectively. The images were deconvoluted by LAS X, lightning module. The refraction index was set to 1.52 for processing. 20–25 planes were collected with a Z-step size of 0.16 µm, and a maximum intensity projection was produced.

## Insulin secretory vesicle enrichment

Following trypsinization, all the steps were performed at 4°C. Cells were gently washed twice with PBS, followed by Dounce homogenization of the cells with 20 strokes in homogenization buffer (HB) (0.3 M sucrose, 10 mM MES, 1 mM EGTA, 1 mM MgSO$_4$, pH 6.3) supplemented with house-made PI cocktail (0.5 M AEBSF, 1 mM E-64, 1.13 mM leupeptin, and 151.36 µM aprotinin). Cell debris was collected by centrifugation at 600 × *g* for 10 min and re-homogenized as described above to lyse the remaining intact cells, followed by a second spin at 600 × *g*. Supernatants were pooled and centrifuged at 5400 × *g* for 15 min to remove mitochondria, ER, and other subcellular compartments of similar density. The pellet was discarded, and the supernatant was centrifuged at 35,000 × *g* for 30 min to sediment insulin vesicles, among other contaminants, yielding up to ~5 µg of dry material per every million cells. This pellet was resuspended in ~450 µL HB and loaded on a density gradient column formed by layering decreasing densities of OptiPrep density media (Sigma-Aldrich D1556) and HB in 0.9 mL fractions of 40%, 35%, 30%, 25%, and 20% OptiPrep in an open-top thin-wall polypropylene tube (Beckman 326819). The density column was then spun in an SW55i Beckman rotor of an ultracentrifuge at 160,000 × *g* for 8 hr to fractionate the insulin vesicle-containing population. Insulin vesicle subpopulations were isolated in fractions of 400 µL, and ELISA and WB were used to identify the fractions most enriched in insulin. A similar dilution was applied to all the fractions. The manufacturer's manual was followed for ELISA (Mercodia 10-1250-01) (*Figure 1—figure supplement 2A*). For WB, fraction samples were mixed with 4X NuPAGE LDS sample buffer (Invitrogen NP0007), loaded on a 15-well NuPAGE 4–12% bis-tris gel (Invitrogen NP0323PK2) and run in a mini-gel tank (Life Technologies A25977) at 200 V for 30 min (Bio-Rad 1645050). Protein was then transferred to a PVDF membrane using iBlot 2 Transfer Stacks (Invitrogen IB24002) in iBlot 2 dry blotting device (Invitrogen IB21001). The membrane was blocked using 5% BSA, then cut according to marker protein size and incubated with antibodies against marker proteins for mitochondria (Cytochrome c Antibody; Novus Biologicals NB100-56503), ER (SEC61B Polyclonal Antibody; Life Technologies PA3015), and insulin vesicles (synaptotagmin IX; Thermo Fisher Scientific PA5-44987; RRID: AB_2610517) at RT for 2–5 hr. Membranes were then washed with 0.1% Tween supplemented PBS (PBST) twice and incubated with the secondary antibody (anti-rabbit IgG, anti-mouse IgG) at RT for 1 hr. Membranes were washed with PBST two to three times and bands were visualized upon addition of SigmaFast BCIP/NBT tablets (Sigma B5655) (*Figure 1—figure supplement 2B*). Fractions containing the highest insulin levels and

high concentration of the vesicle marker synaptotagmin IX were regarded as insulin vesicle samples and were further tested for DLS using Wyatt Technology's Mobius to confirm the size distribution of particles corresponding to the insulin vesicle diameter, reportedly 200–500 nm. Data was analyzed in DYNAMICS and manually corrected against the control (HB) (*Figure 1—figure supplement 2C*). Insulin vesicle fractions confirmed to have the expected insulin vesicle diameter by DLS were then pooled together and spun down at 35,000 × *g* for 15 min to sediment the insulin vesicles. The pellet obtained from this step was used for immunolabeling of insulin vesicles.

## Verification of intact isolated insulin vesicles

Fluorescent confocal microscopy was used to verify that enriched insulin vesicles were intact. Suspended insulin vesicles were imaged using a 63× water objective on 0.6-mm-thick coverslips on a Mica (Leica Microsystems). The signal was collected with an excitation wavelength of 488 nm and emission wavelength of 509 nm. Images were deconvoluted by LAS X, lightning module. The refractive index was set to 1.33 for processing.

Negative stain EM was used to evaluate if the isolated insulin vesicles were intact prior to analysis with DC-iDEP. In this case, we used a modified INS-1 cell line that expresses a human insulin and green fluorescent protein-tagged C peptide (hPro-CpepSfGFP) (*Haataja et al., 2013*) (GRINCH cells) which allowed for quick visual verification of isolated insulin-containing vesicles. Insulin vesicles isolated from GRINCH cells (RRID:CVCL_WH61) using an EDTA-supplemented enrichment strategy were deposited directly on EM grids (Formvar/Carbon 200 mesh, Copper, Ted Pella). Insulin vesicles were allowed to settle on the grid for 10 min at 20°C before blotting. A tungsten-based negative stain (Nano-W, Ted Pella) was spotted onto the grid, incubated at 20°C for 30 s, then excess stain was blotted. The negative staining was repeated twice more, then the grid was allowed to dry at 37°C for 1 hr. Prepared grids were imaged at RT on a Talos F200C G2 Transmission Electron Microscope (Thermo Fisher Scientific) running at 80 kV acceleration voltage.

To further validate the integrity of the insulin vesicles isolated, they were applied onto EM grids (Ted Pella Lacey Carbon, 200 mesh, TH, Gold) at 4°C and 100% humidity for cryo-EM imaging. Excess buffer was blotted for 2 s before grids were plunge-frozen in liquid ethane using a Vitrobot Mark IV (Thermo Fisher Scientific). Grids were imaged under cryogenic conditions using a 200 kV Glacios Cryo TEM equipped with a Falcon 4 detector (Thermo Fisher Scientific) or a 300 kV Krios G3i equipped with a Gatan K3 direct detection camera (Thermo Fisher Scientific).

## Immunolabeling of insulin vesicles

The pellet was resuspended in HB and incubated with 5–10 μg of anti-synaptotagmin IX (Thermo Fisher Scientific PA5-44987; RRID:AB_2610517; 1:50–1:100) overnight to tag an insulin vesicle marker. This primary antibody was then fluorescently labeled by 2–4 hr incubation with 5–10 μg of Alexa 568-conjugated secondary antibody (Invitrogen A-11011; RRID:AB_143157; 1:50–1:100) (*Figure 1*). Insulin vesicles were washed with HB three times to remove the excess antibody and were finally resuspended in a low conductivity buffer (LCHB; 0.3 M sucrose, 5 mM MES, pH 6.3), which is compatible with dielectrophoresis studies.

## Device fabrication

The design and fabrication methods of the separation device were described in prior publications (*Greider et al., 1969*). The device contains a 27-gate sawtooth channel with a depth of approximately 20 μm and a length of 3.5 cm from inlet to outlet (*Figure 1C*). The distance between two paired triangle tips (gates) decreases from 73 to 25 μm in the channel. The gate size decreases approximately 5 μm after every three repeats. Direct current was applied to the device between the inlet and outlet. The potentials were between 0 and 1800 V for testing.

The microfluidic devices were fabricated by standard soft lithographic technique as described previously (*Staton et al., 2010*). The design of the channel was created by AutoCAD (Autodesk, Inc, San Rafael, CA, USA) and was used for fabricating a photomask. The channel was created by exposing AZ P4620-positive photoresist (AZ Electronic Materials, Branchburg, NJ, USA) on Si wafer CEM388SS (Shin-Etsu MicroSi, Inc, Phoenix, AZ, USA) by contact lithography. Extra materials were removed from the Si wafer. A weight of 22 g of polydimethylsiloxane (PDMS, Sylgard 184, Dow/Corning, Midland, MI, USA) was used to fabricate four channels simultaneously. The PDMS mixture was placed on the

Si wafer template and left to stand for 30 min to allow bubbles to dissipate, and then it was baked for 1 hr at a temperature of 80°C. Holes 2.5 mm in diameter were punched for inlet and outlet reservoirs. Each channel was capped with a glass microscope slide to fabricate the enclosed channels after cleaning and activation by plasma cleaner (Harrick Plasma, Ithaca, NY, USA) with a voltage of 50 kV.

### Electric field simulations

Finite element modeling (COMSOL, Inc, Burlington, MA, USA) of the distribution of the electric field in the microchannel was performed as previously detailed (*Staton et al., 2010*). The *AC/DC module* was used to interrogate the $\vec{E}$, $\nabla\left|\vec{E}\right|^2$, and $\frac{\nabla\left|\vec{E}\right|}{E^2}\cdot\vec{E}$ in an accurately scaled 2D model of the microchannel.

### DC-iDEP

The separation channel was treated with 5% (wt/vol) BSA for 15 min followed by a wash with LCHB. A volume of 15 µL of the insulin vesicle sample was introduced to the device from the inlet. This sample fraction contains particles, ~75% of which have radii characteristic of insulin vesicle as apparent from DLS experiments (*Figure 1—figure supplement 2C*). The volume in each reservoir was maintained with LCHB to prevent pressure-induced flow. Direct current was applied at 600, 900, 1200, 1500, and 1800 V between the inlet and outlet and particles were driven through the channel by the EK force experienced.

### Imaging of vesicles during separation

Images and recordings were acquired using an Olympus IX70 inverted microscope with 4×, NA 0.16, and 20×, NA 0.40, objectives. The 20× objective was used to inspect the channel to confirm that the device was properly formed. The 4× objective was used in recording the intensity of the insulin vesicle signal shown in *Figure 1*. A mercury short arc lamp (H30102 w/2, OSRAM) and a triple band pass cube (Olympus, Center Valley, PA, USA) were used for sample illumination and detection (excitation: 400/15-495/15-570/25 nm; dichroic: 410-510-590 nm; emission: 460/20-530/30-625/50 nm). Fluorescent intensities of immunolabeled insulin vesicles were recorded using the 4× objective by a LightWise Allegro camera (LW-AL-CMV12000, USB3, 0059-0737-B, Imaging Solutions Group) after the voltage had been applied 90 s. Images were recorded from three to four biological replicates at each gate (27 total gates) for any given voltage. Images were further processed in ImageJ (NIH, freeware). The intensity at each gate was recorded along with the intensity of a nearby open area of the channel (image intensity background), which was subtracted from the intensity at each gate to adjust for any variation in illumination intensity. The data for each applied voltage value was normalized to the highest intensity within that dataset.

### Theory

The forces exerted on bioparticles in the microfluidic device in the presence of direct current include DEP and EK forces. Separation of subpopulations is achieved based on the different magnitude of the forces each bioparticle experiences, related to the properties of the particles, including their radius, conductivity, and zeta potential.

The EK mobility, $\mu_{EK}$, and velocity, $\vec{v}_{EK}$, are described as:

$$\mu_{EK} = \mu_{EP} + \mu_{EOF}$$

$$v_{EK} = \mu_{EK}\vec{E}$$

where $\mu_{EP}$ is the electrophoretic (EP) mobility and $\mu_{EOF}$ is the electro-osmotic flow mobility. The DEP mobility, $\mu_{DEP}$, and velocity, $\vec{v}_{DEP}$, can be expressed as:

$$\mu_{DEP} = \frac{\varepsilon_m r^2 f_{CM}}{3\eta}$$

$$\vec{v}_{DEP} = \mu_{DEP}\nabla\left|\vec{E}\right|^2$$

where $r$ is the radius of the particle, $f_{CM}$ is the Clausius-Mossotti factor, $\varepsilon_m$ is the dielectric constant of the solution, and $\eta$ is the viscosity.

A combination of biophysical properties of the particles, such as insulin vesicles, determines the location where they will be captured in a microfluidic device. Capture occurs when the EK velocity of the particle is equal to that of DEP. The condition is:

$$\vec{j} \cdot \vec{E} = 0$$

$$\frac{\nabla \left| \vec{E} \right|^2}{E^2} \cdot \vec{E} \geq \frac{\mu_{EK}}{\mu_{DEP}}$$

where $\vec{E}$ is the electric field intensity, $\vec{j}$ is the particle flux, and $\nabla|\vec{E}|$ is the gradient of the electric field. The ratio of EK to DEP mobilities (EKMr, $\frac{\mu_{EK}}{\mu_{DEP}}$) can be used to characterize the biophysical properties of different subpopulations (*Liu et al., 2019*). The EKMr of an insulin vesicle is larger than $\frac{\nabla|\vec{E}|^2}{E^2} \cdot \vec{E}_1$ of the gates which it has passed through and is smaller/equal to $\frac{\nabla|\vec{E}|^2}{E^2} \cdot \vec{E}_2$ of the gates where it is captured. In this way, insulin vesicles are separated in the microfluidic channel, thus measuring the EKMr values for the insulin vesicles.

# Acknowledgements

The authors would like to thank Raymond Stevens and the Bridge Institute for supporting our project and members of the Pancreatic β-Cell Consortium for their inspiring discussions and feedback. We additionally thank Chris Hanson for assisting with the cell culture, Yekaterina Kadyshevskaya for helping with illustrations, Claire Cato for feedback on the manuscript, Brett Barbaro for assisting with analysis of the existing proteomics data, the USC NanoBiophysics Core Facility for facilitating the DLS experiments, and the USC Center of Excellence in Nano Imaging for facilitating the EM. This material is based upon work supported by the National Science Foundation Graduate Research Fellowship Program under Grant No. DGE-1842487. Any opinions, findings, and conclusions or recommendations expressed in this material are those of the author(s) and do not necessarily reflect the views of the National Science Foundation. Research reported in this publication was supported by the National Institute of General Medical Sciences of the National Institutes of Health under award number R35GM154893.

# Additional information

### Competing interests
Mark A Hayes: serves as the CSO of CBio, and COB, CEO & CSO of Hayes Diagnostics, Inc. The other authors declare that no competing interests exist.

### Funding

| Funder | Grant reference number | Author |
|---|---|---|
| National Institutes of Health | 5R03AI133397-02 | Mark A Hayes |
| National Science Foundation | DGE-1842487 | Ashley Archambeau |
| National Institutes of Health | R35GM154893 | Kate L White |

The funders had no role in study design, data collection and interpretation, or the decision to submit the work for publication.

### Author contributions
Mahta Barekatain, Data curation, Formal analysis, Investigation, Writing – original draft, Writing – review and editing; Yameng Liu, Ashley Archambeau, Formal analysis, Investigation, Writing – original draft, Writing – review and editing; Vadim Cherezov, Formal analysis, Supervision, Investigation, Writing – original draft, Writing – review and editing; Scott Fraser, Conceptualization, Supervision, Investigation, Writing – original draft, Writing – review and editing; Kate L White, Conceptualization,

Formal analysis, Supervision, Writing – original draft, Project administration, Writing – review and editing; Mark A Hayes, Conceptualization, Formal analysis, Supervision, Funding acquisition, Methodology, Writing – original draft, Project administration, Writing – review and editing

### Author ORCIDs
Ashley Archambeau (iD) http://orcid.org/0000-0002-9933-1787
Vadim Cherezov (iD) https://orcid.org/0000-0002-5265-3914
Kate L White (iD) https://orcid.org/0000-0001-8894-9621

### Decision letter and Author response
Decision letter https://doi.org/10.7554/eLife.74989.sa1
Author response https://doi.org/10.7554/eLife.74989.sa2

## Additional files

### Supplementary files
• MDAR checklist

### Data availability
All data generated or analysed during this study are included in the manuscript and supporting file; Source Data files have been provided for Figures S2 and 4.

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
