## [Editor Report]

This important study presents an exciting new method for separating organelles in an unbiased way and applies this method to the separation of distinct subpopulations of insulin vesicles. Solid evidence is presented that this method is capable of separating distinct subpopulations of insulin vesicles, but the identification of these subpopulations is incomplete and the biological significance of the proposed changes in vesicle populations remains unclear. This work will be of interest to cell biologists studying a variety of organelles.

---

## [Decision Letter]

**Decision letter after peer review:**

Thank you for submitting your article "iDEP-assisted isolation of insulin secretory vesicles" for consideration by *eLife*. Your article has been reviewed by 4 peer reviewers, including Michael Ailion as the Reviewing Editor and Reviewer #1, and the evaluation has been overseen by Nancy Carrasco as the Senior Editor. The following individuals involved in review of your submission have agreed to reveal their identity: Alex J B Kreutzberger (Reviewer #2).

All of the reviewers are excited about the new technique for separating insulin vesicles, but there was also a concern that the manuscript falls short in definitively demonstrating that the vesicles being measured are indeed insulin vesicles, with the biological significance of the proposed changes in vesicle populations remaining unclear. It is also unclear so far whether the technique permits further biochemical characterization of these vesicles. To be published in *eLife*, these issues must be resolved. Essential revisions are listed below, followed by a general evaluation summary, and the individual reviews of each of the reviewers.

Essential revisions (for the authors):

1. What are the vesicles being measured? Are these insulin containing vesicles? There is concern given that these vesicles are identified by a single marker and that the data show poor overlap of this marker with insulin by immunostaining. It needs to be resolved which specific synaptotagmin is labeled and further evidence that these vesicles are insulin vesicles needs to be provided. This could possibly be addressed by the use of additional markers and some type of composition or visual (EM) identification.

2. What do the changes in vesicle populations mean? Are the differences biologically meaningful? Without knowing what the various populations of vesicles are, it is difficult to understand what the changes in response to glucose mean. Given that β cells only secrete 5-10% of their content with glucose stimulation, it is hard to imagine that whole pools of vesicles change. If they do, this would be fascinating and a significant finding, but parameters need to be defined that could account for the magnitude of the observed changes. For example, how much remodeling of proteins or lipids would be required to account for the observed differences? Is there a way to provide some sort of standardization or comparison for such changes?

3. One of the major advances of the new technology presented is the claimed ability to isolate populations of vesicles for further characterization. However, the feasibility of this is unclear and it is not demonstrated in the paper. Evidence needs to be provided that this method will yield enough sample material to enable further biochemical characterization. This could be as simple as taking two of the different granule populations separated using iDEP and running a Western blot for a few markers of insulin vesicles. That would demonstrate that the method is sorting insulin vesicles and that after sorting they can be characterized to some degree.

*Reviewer #1 (Recommendations for the authors):*

1. In multiple places, the paper refers to performing experiments on β cells (e.g. lines 17, 258, 275, 366, 371). INS-1E cells are not true β cells and should not be referred to as such.

2. There should be more detailed methods describing the immunolabeling of the vesicles. What pellet is referred to in line 166? How much primary and secondary antibody were used and how long was staining performed?

3. t tests are used to compare specific data points between low and high glucose conditions (Figure 4), but such comparisons require a statistical test that corrects for multiple comparisons.

4. The Results sections describing the data in Figure 4 were difficult to read. Rather than using phrases like "pattern with discernible features," it would help to be specific about what features are being referred to.

5. line 193: what is meant by "Behavior comparison?" This is an unintuitive heading.

6. Line 175 should refer to Figure 1C (not Figure 2C).

7. The transparent reporting form does not point to lines in the manuscript that specify how the sample size was determined, and I could not find that information in the manuscript. Other lines referred to in the transparent reporting form are off and appear to have shifted.

8. According to *eLife* instructions, authors should avoid acronyms in the title:

Titles of *eLife* research papers should avoid unfamiliar abbreviations or acronyms, or authors should spell out in full or provide a brief explanation for any acronyms. Please revise your title with this advice in mind.

9. For *eLife* papers, the biological system should be indicated in the title and/or abstract:

The title and/or abstract should provide a clear indication of the biological system under investigation (i.e., species name or broader taxonomic group, if appropriate). For this paper, the biological system would be INS-1E insulinoma cells. Please revise your title and/or abstract with this advice in mind.

*Reviewer #2 (Recommendations for the authors):*

1. Synaptotagmin V was chosen as a marker for insulin granules before sorting subpopulations. Synaptotagmin isoforms are well known to differentially sort into sub-populations (Zhang et al. 2011 Mol. Biol Cell, Rao et al. 2014 Mol Biol Cell, Rao et al. 2017 JGP, Kreutzberger et al. 2020 *eLife*). Could you be missing populations because Synaptotagmin V is only on a subset of insulin granules (ie are you sorting a sub-population of a sub-population)? Synaptotagmin VII is well characterized to control a distinct pool of insulin granules – would granules within this population be missed in your isolation procedure?

2. Insulin granules were found to have different biophysical properties when cells were treated with glucose. Why is this a significant finding?

This glucose stimulation will trigger secretion and release of insulin and other secretory properties from some fraction of the insulin granules. Would you not expect the loss of these secretory contents to create a difference in dielectrophoretic and electrokinetic properties of the granules by no longer having the released secretory content?

3. If there was more of a biochemical characterization of what compositional differences in lipids and proteins are present within isolated populations of granules – the impact of this work would be greatly increased.

*Reviewer #3 (Recommendations for the authors):*

1) Studies with additional stimuli or cell treatments may reveal some clues regarding the identities of the major populations being observed that are changing.

2) Control experiments modifying vesicle parameters would be really helpful to understand the implications. This could be done chemically or perhaps knocking down certain vesicle components targeting features of trafficking or maturation.

3) Perhaps examining maturation markers, or age markers, or at least various insulin vesicle markers would be informative and or further confirmation of the changes in response to glucose (i.e. are general to the insulin population and not specific to SytV-positive vesicles).

4) Examination of insulin vesicles from primary cells (rather than an insulinoma cell line), although challenging due to the limited numbers of cells, would greatly improve the impact of this study. Alternatively, data from other insulinoma cell lines may be a useful comparison to determine if the changes are universal or specific to the INS-1E cell line.

---

## [Author Response]

Essential revisions (for the authors):Reviewer #1 (Recommendations for the authors):1. In multiple places, the paper refers to performing experiments on β cells (e.g. lines 17, 258, 275, 366, 371). INS-1E cells are not true β cells and should not be referred to as such.

We thank the reviewer for this comment. The incorrect labeling of INS-1E cells as β cells has been fixed to refer to them as either “INS-1E” or “INS-1E insulinoma” cells.

2. There should be more detailed methods describing the immunolabeling of the vesicles. What pellet is referred to in line 166? How much primary and secondary antibody were used and how long was staining performed?

We have modified the methods section to clarify which pellet is being referred to.

“The pellet obtained from this step was used for immunolabeling of insulin vesicles.” page 12, lines 309-310.

We have also added the amounts of antibody used in each step.

“The pellet was resuspended in HB and incubated with 5-10 μg of anti-synaptotagmin IX (Thermo Fisher Scientific PA5-44987; RRID: AB_2610517; 1:50-1:100) overnight to tag an insulin vesicle marker. This primary antibody was then fluorescently labeled by 2-4 hours incubation with 5-10 μg of Alexa 568-conjugated secondary antibody (Invitrogen A-11011; RRID: AB_143157; 1:50-1:100) (Figure 1). Insulin vesicles were washed with HB three times to remove the excess antibody and were finally resuspended in a low conductivity buffer (LCHB; 0.3 M sucrose, 5 mM MES, pH 6.3), which is compatible with dielectrophoresis studies.” page 13, line 334-339.

3. t tests are used to compare specific data points between low and high glucose conditions (Figure 4), but such comparisons require a statistical test that corrects for multiple comparisons.

We thank the reviewer for pointing this out. We have performed new statistical analysis using an ANOVA test with the Bonferroni post hoc correction. We have also rewritten this section to explain the data and our interpretation in a more straightforward manner as requested by major comment # 3 (page 7, lines 158-165 of the manuscript).

4. The Results sections describing the data in Figure 4 were difficult to read. Rather than using phrases like "pattern with discernible features," it would help to be specific about what features are being referred to.

We thank the reviewer for the helpful feedback and have rewritten this section as addressed in major comment #3. We also made minor edits to all Results sections which describe figure 4 to improve clarity.

5. line 193: what is meant by "Behavior comparison?" This is an unintuitive heading.

We appreciate this concern and have changed it to the heading below:

“Comparison of n- and g-insulin vesicle separation patterns within the iDEP device” page 6, line 157

6. Line 175 should refer to Figure 1C (not Figure 2C).

We thank the reviewer and have fixed the error.

7. The transparent reporting form does not point to lines in the manuscript that specify how the sample size was determined, and I could not find that information in the manuscript. Other lines referred to in the transparent reporting form are off and appear to have shifted.

We thank the reviewer for bringing this to our attention. We have added a section addressing sample size determination:

“No statistical methods were used to predetermine sample size. The experiments were not randomized, and investigators were not blinded to allocation during experiments and outcome assessment.” Page 10, lines 237-238.

8. According to eLife instructions, authors should avoid acronyms in the title:Titles of eLife research papers should avoid unfamiliar abbreviations or acronyms, or authors should spell out in full or provide a brief explanation for any acronyms. Please revise your title with this advice in mind.

We thank the reviewer for pointing this out and have made the correction below.

“Insulator-based dielectrophoresis-assisted separation of insulin secretory vesicles”

9. For eLife papers, the biological system should be indicated in the title and/or abstract:The title and/or abstract should provide a clear indication of the biological system under investigation (i.e., species name or broader taxonomic group, if appropriate). For this paper, the biological system would be INS-1E insulinoma cells. Please revise your title and/or abstract with this advice in mind.

We thank the reviewer for this comment and have made the appropriate changes to our abstract.

“Organelle heterogeneity and inter-organelle contacts within a single cell contribute to the limited sensitivity of current organelle separation techniques, thus hindering organelle subpopulation characterization. Here we use direct current insulator-based dielectrophoresis (DC-iDEP) as an unbiased separation method and demonstrate its capability by identifying distinct distribution patterns of insulin vesicles from INS-1E insulinoma cells. A multiple voltage DC-iDEP strategy with increased range and sensitivity has been applied, and a differentiation factor (ratio of electrokinetic to dielectrophoretic mobility) has been used to characterize features of insulin vesicle distribution patterns. We observed a significant difference in the distribution pattern of insulin vesicles isolated from glucose-stimulated cells relative to unstimulated cells, in accordance with maturation of vesicles upon glucose stimulation. We interpret the difference in distribution pattern to be indicative of high-resolution separation of vesicle subpopulations. DC-iDEP provides a path for future characterization of subtle biochemical differences of organelle subpopulations within any biological system.” Page 1, line 17

Reviewer #2 (Recommendations for the authors):1. Synaptotagmin V was chosen as a marker for insulin granules before sorting subpopulations. Synaptotagmin isoforms are well known to differentially sort into sub-populations (Zhang et al. 2011 Mol. Biol Cell, Rao et al. 2014 Mol Biol Cell, Rao et al. 2017 JGP, Kreutzberger et al. 2020 eLife). Could you be missing populations because Synaptotagmin V is only on a subset of insulin granules (ie are you sorting a sub-population of a sub-population)? Synaptotagmin VII is well characterized to control a distinct pool of insulin granules – would granules within this population be missed in your isolation procedure?

We thank the reviewer for their insight. As the antibody we used to label insulin vesicles is likely targeting only a subset of insulin vesicles (Synaptotagmin IX-positive insulin vesicles), our visualization and analysis are limited to this subset and do not include Synaptotagmin VII-positive insulin vesicles. Nonetheless, our results demonstrated the potential for iDEP to reveal heterogeneity in supposedly similar particles. We have begun experiments using a modified INS-1 cell line with a GFP-tagged C-peptide (hPro-CpepSfGFP, GRINCH cells RRID:CVCL_WH61). This cell line would allow for the detection of a more complete insulin vesicle population with iDEP.

2. Insulin granules were found to have different biophysical properties when cells were treated with glucose. Why is this a significant finding?This glucose stimulation will trigger secretion and release of insulin and other secretory properties from some fraction of the insulin granules. Would you not expect the loss of these secretory contents to create a difference in dielectrophoretic and electrokinetic properties of the granules by no longer having the released secretory content?

We appreciate this comment. The observed differences between n- and g- glucose are not a novel finding but rather the first demonstration of the detection of these changes with the iDEP device. There is likely a balance of some subpopulations being absent due to secretion and new populations emerging due to biochemical maturation of insulin and vesicles. This method has the potential to be further developed for isolation of these subpopulations for more thorough characterization. As mentioned for Review #1 question #1, we have expanded our discussion to include possible explanations of the observed changes in the subpopulations between treatment conditions (Page 7-8, lines 176-191).

3. If there was more of a biochemical characterization of what compositional differences in lipids and proteins are present within isolated populations of granules – the impact of this work would be greatly increased.

We agree with the reviewer that further biochemical characterization of these subpopulations would be useful. However, it is important to note that the capability of this method to isolate the separated and concentrated subpopulations is still under development. Additional method refinement is necessary and unfortunately, completing it in a timely manner for resubmission of this paper will not be feasible.

Reviewer #3 (Recommendations for the authors):1) Studies with additional stimuli or cell treatments may reveal some clues regarding the identities of the major populations being observed that are changing.

We thank the reviewer for this suggestion. We are planning experiments to further probe these subpopulations in our future studies. We expect to see some subpopulations that have been described in the literature, some combinations of these subpopulations, and potentially some subpopulations that have not been previously reported. We also plan to stimulate cells to gain a better understanding of how these subpopulations behave under different conditions. We view our current work as a method development proof-of-concept and plan to apply it for more detailed biological exploration in future studies.

2) Control experiments modifying vesicle parameters would be really helpful to understand the implications. This could be done chemically or perhaps knocking down certain vesicle components targeting features of trafficking or maturation.

We agree this would be a useful approach for future studies to dissect the biological roles for different subpopulations. However, we believe this is out of scope for our current manuscript.

3) Perhaps examining maturation markers, or age markers, or at least various insulin vesicle markers would be informative and or further confirmation of the changes in response to glucose (i.e. are general to the insulin population and not specific to SytV-positive vesicles).

We agree with the reviewer that these experiments would be informative on the biology, but we believe these experiments are out of scope for our current study. We plan to conduct experiments with the GRINCH cell line to better visualize these changes with a more complete set of insulin vesicles. We also plan to explore the age and maturation of each subpopulation and how those factors fluctuate under different conditions. These significant biology efforts are more suitable for separate publications.

4) Examination of insulin vesicles from primary cells (rather than an insulinoma cell line), although challenging due to the limited numbers of cells, would greatly improve the impact of this study. Alternatively, data from other insulinoma cell lines may be a useful comparison to determine if the changes are universal or specific to the INS-1E cell line.

We agree that studying insulin vesicles from primary cells would yield valuable data, and we plan to use primary cells to compare results in future studies. We have observed similar results from insulin vesicles isolated from the GRINCH cell line, and plan to extend our experiments to other cell lines which include genetic knockouts of proteins involved in insulin maturation.